# Simulation analysis of fertilizer discharge process using the Discrete Element Method (DEM)

**Kemoh Bangura**[1,2], **Hao Gong**[1], **Ruoling Deng**[1], **Ming Tao**[1], **Chuang Liu**[1], **Yinghu Cai**[1], **Kaifeng Liao**[1], **Jinwei Liu**[1], **Long Qi**[1] *

**1** College of Engineering, South China Agricultural University, Guangzhou, China, **2** Rokupr Agricultural Research Center (RARC), Sierra Leone Agricultural Research Institute (SLARI), Freetown, Sierra Leone

* qilong@scau.edu.cn

**Data Availability Statement:** All relevant data are within the manuscript and its Supporting files.

**Funding:** The research was funded by the National Key R&D Program of China, grant number 2018

## Abstract

Fertilizer discharge process is a critical part of fertilizer application, as it affects the fertilizer discharge rate and uniformity of fertilizer application. In this study, a spiral grooved-wheel fertilizer discharge device was designed to replace the conventional straight grooved-wheel. Comparisons of the fertilizer discharge performance of the two grooved-wheel types were performed through tests and simulations using the discrete element method (DEM). The discharge performance of the two discharge devices was assessed by measuring the discharge mass rate, discharge uniformity, and the falling velocity of the fertilizer particles. Results showed that under similar conditions, the fertilizer discharge mass rate of the spiral grooved-wheel was higher than that of the straight grooved-wheel. The fertilizer discharge uniformity of the spiral grooved-wheel was much better than that of the straight grooved-wheel. The average falling velocity of fertilizer particles through the discharge spout was higher under the spiral grooved-wheel. The relative errors between the test and simulation results for the discharge mass rates, discharge uniformity, and particle falling velocities of the spiral grooved-wheel were all less than 10%. The developed spiral grooved-wheel exhibited a better performance than the conventional straight grooved-wheel, in all the aspects examined. The results serve as a theoretical basis for guiding the design of high-performance fertilizer applicators.

## Introduction

As human demand for food increases, the application of fertilizer has become an important means to increase grain yield in the world. Currently, China has one of the highest rates of chemical fertilizer use, with fertilizer being applied to 67% of grain crops [1]. Therefore, how to improve fertilizer use efficiency has become a national and global concern [2–4]. One of the ways to improve the efficiency of using chemical fertilizers is through the development and promotion of improved and sustainable fertilizer discharge devices to replace the conventional fertilizer application methods for better uniformity of fertilizer application. Uniformity of

YFD0200303, the Natural Science Foundation of China, grant number 51875217, the key R&D Program of Guangdong, grant number 2019B020221003, the Science Foundation of Guangdong for Distinguished Young Scholars, grant number 2019B151502056, the Earmarked Fund for Modern Agro-industry Technology Research System, grant number CARS-01-43 (L. Q).

**Competing interests:** The authors have declared that no computing interest exist.

fertilizer application refers to the degree of uniform distribution of the fertilizer discharge from the fertilizer applicator during operation. Uniformity allows for better distribution of the fertilizer in the root zone and better nutrient utilization efficiency.

In recent years, several theoretical and experimental studies have been performed on fertilizer discharge devices [1, 5, 6]. Among those developed in China, the straight grooved-wheel was the most widely used for the discharge of fertilizer because of its advantages, such as simple structure, convenient manufacturing and processing, and good versatility. However, there are several challenging issues for the straight grooved-wheel, such as poor stability during operation, low discharge mass rate, and poor fertilizer discharge uniformity [7–9]. To overcome these challenges, this study compared the straight grooved-wheel with a spiral grooved-wheel. The spiral grooved-wheel has been reported for producing good results on fertilizer discharge mass rate and uniformity [5, 10]. In addition, the spiral grooved-wheel can handle various shapes and sizes of fertilizer granules [5, 11, 12]. In these existing studies, there was a lack of comparison between the spiral grooved-wheel and the straight grooved-wheel on fertilizer discharge performance. This study aimed to bridge this gap.

Understanding fertilizer discharge characteristics using experiments is difficult because the fertilizer discharge process is complex as it deals with the movement of particles, and the interactions of particles and machine. Computer simulation using discrete element method (DEM) would be a good approach in understanding and observing the microscopic interaction between particles and against the machine. The DEM was considered to be an effective simulation tool in dealing with particulate materials and has been applied in many fields in agriculture [13, 14]. In recent years, granular fertilizers have been simulated using DEM. Lv et al. [1] used DEM to simulate the discharge process of fertilizer spreaders having an outer groove wheel. The DEM simulation results were compared with test results, and the preliminary results indicated the validity and effectiveness of the DEM simulation. Ding et al. [6] developed a dual-band fertilizer applicator to simultaneously deliver starter and base fertilizer into the soil. The performance of the applicator was modeled using the DEM. The results showed that the DEM model was able to simulate the dual banding fertilizer application with a reasonably good accuracy. Yinyan et al. [7] studied the spreading performance of a centrifugal variable-rate fertilizer applicator by conducting DEM simulation tests. The results showed that the coefficient of variation for the developed variable-rate spreader was reduced. Results from these studies demonstrated that the DEM is an effective tool to simulate granular materials. Thus, the DEM was used in this study to simulate the fertilizer discharge performance of two grooved-wheels, the spiral grooved-wheel and the straight grooved-wheel. The objective was to study the influence of the spiral grooved-wheel and the straight grooved-wheel on fertilizer discharge mass rate, discharge uniformity, and particle falling velocity at different wheel speeds.

## Materials and methods

### Design of the fertilizer discharge device

The fertilizer discharge model designed consisted of a fertilizer box, a spiral grooved-wheel, a discharge box, a discharge shaft, and a fertilizer discharge spout (Fig 1). The height of the fertilizer discharge model was kept moderate at 240 mm. This height ensures that its center of gravity after being filled with fertilizer will not be too high, and this will prevent the fertilizer particles from falling [15]. The working diameter of existing grooved-wheels was in the range of 50 to 65 mm and the grooved-wheel length generally varied from 25 to 50 mm [5, 16, 17]. Based on this, the working diameter and length of the spiral grooved-wheel designed in this study were 53 and 33 mm, respectively. In the literature, the range of the groove radius was

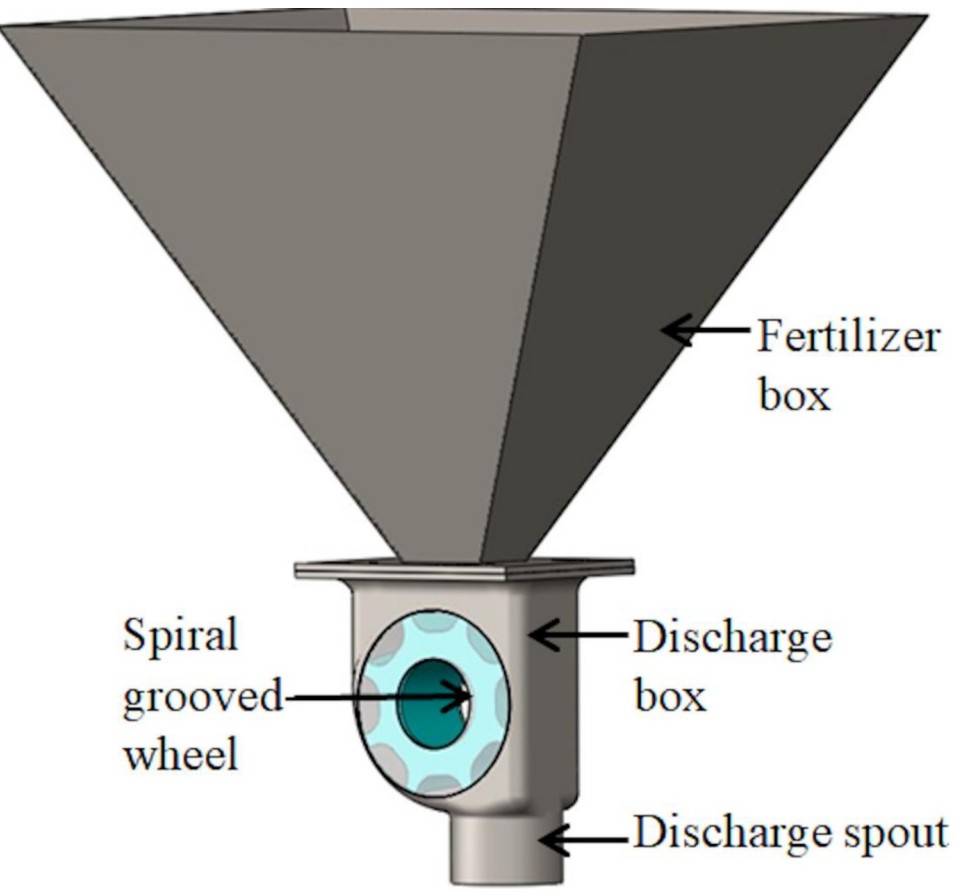

**Fig 1. Spiral grooved-wheel fertilizer applicator.**

generally 2 to 9 mm, and the number of grooves varied generally from 6 to 10 [5, 16]. Within these ranges, a groove radius (r) of 6 mm and the number of grooves (z) of 8 were selected. Fig 2 shows the structural dimensions of the fertilizer discharge wheel used in this study.

The fertilizer box was designed as a hopper shape, which had a rectangular-shaped cross-section at the top (200 × 200 mm) and at the bottom (74 x 72 mm) (Fig 3). The inclination angle of the bottom plate of the fertilizer box was 60°. This angle ensured that the fertilizer could flow into the grooved wheel smoothly [5, 18].

## Model development

**Fertilizer particle model.** In this study, common compound and Urea fertilizers were used. For each fertilizer, 100 fertilizer granules were randomly selected from a bulk fertilizer sample. The length, width, and thickness of fertilizer granules were measured with a caliper, and the equivalent diameter and sphericity were calculated using the following equations:

$$D = \sqrt[3]{LWT} \tag{1}$$

$$\phi = \frac{D}{L} \tag{2}$$

where $L$, $W$, and $T$ are the length, width, and thickness of fertilizer granule, respectively (mm);

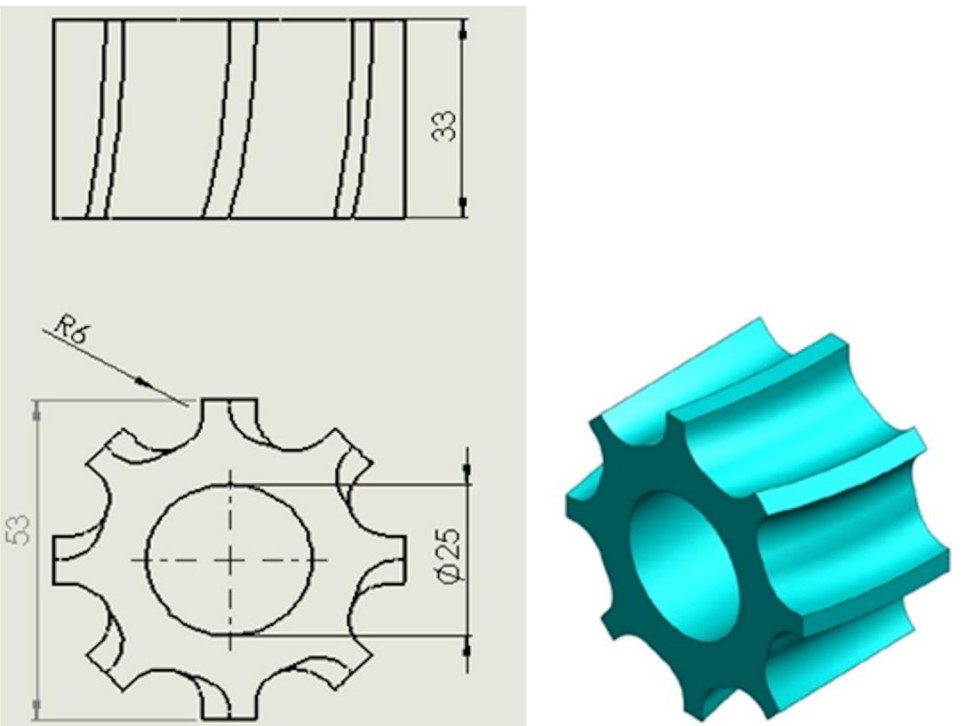

**Fig 2. Main structural dimensions and model of the spiral grooved-wheel.**

$D$ is the equivalent diameter of the granule (mm); and $\phi$ is the sphericity of the granule (%) [19, 20]. The results are presented in Table 1.

The results in Table 1 showed that the differences in particle length, width, and thickness of the two fertilizers were relatively small, and the sphericity was greater than 90% for both fertilizers. Therefore, they were simplified as spheres in the simulation. Spherical particles have also been previously used to simulate fertilizer granules [1, 6, 21]. The diameter of the model fertilizer particles was set as the equivalent diameter shown in Table 1.

**Model parameters.**   Model inputs include parameters for fertilizer, machine, and contacts. The parameters of each material were density, Poisson's ratio, and shear modulus. The parameters of each contact were coefficient of restitution, static friction coefficient, and rolling friction coefficient. Among these parameters, the particle density of the fertilizer was measured through experiments, and the rest of the parameters were taken from different works of literature [5, 6, 22, 23]. The model parameters are summarized in Tables 2 and 3.

## Model validation tests

**Description of the testing equipment.**   To validate model simulation results, two fertilizer discharge devices, spiral and straight grooved-wheels, were tested. Both devices had the same configuration as shown in Figs 2 and 3, except for the grooved-wheel. The test set up (Fig 4(A) and 4(B)) consisted of the discharge unit, a DC motor, a speed control unit, a high-speed phantom camera, and a camera control computer. In the tests, the speed of the discharge wheel was set at a constant speed of 37.5 rpm. This speed was selected on the basis that fertilizer discharge units with grooved-wheels work well between 30 to 45 [1, 6, 24]. The grooved-wheel was driven by the DC motor and the speed could be adjusted by the speed control system. The resultant fertilizer discharge mass rate, discharge uniformity, and particle falling velocity were measured as described below, and the data were used to compare with simulated values.

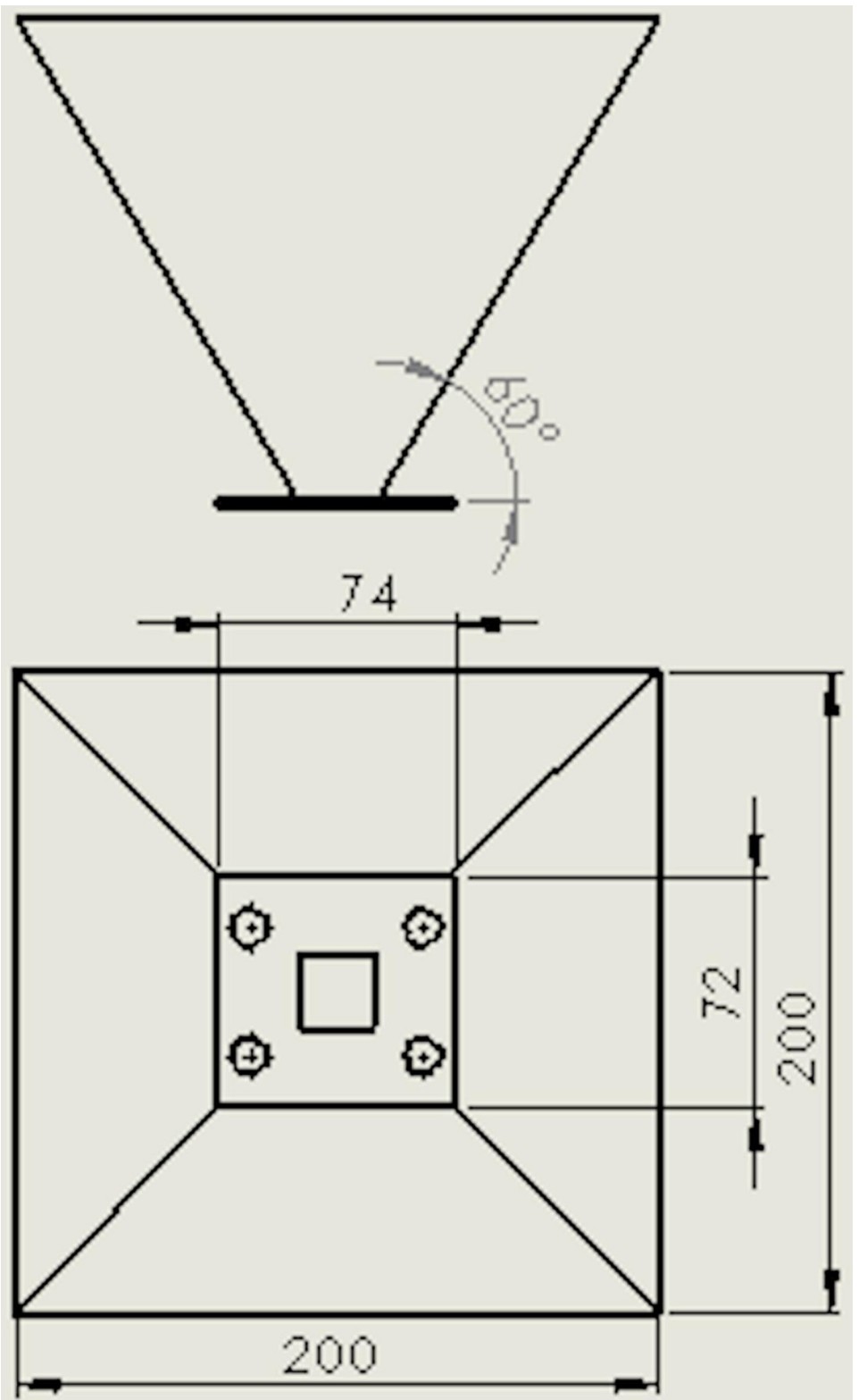

**Fig 3. Dimensions of the fertilizer box.**

**Table 1. Triaxial size, equivalent diameters, and sphericities of fertilizer granules.**

| Fertilizer type | Length (mm) | Width (mm) | Thickness (mm) | Equivalent diameter (mm) | Sphericity (%) |
|---|---|---|---|---|---|
| Compound fertilizer | 4.60 | 4.11 | 3.77 | 4.13 | 90.71 |
| Urea fertilizer | 4.01 | 3.74 | 3.57 | 3.77 | 94.13 |

**Measurements.** Tests were replicated five times for each grooved-wheel, and each test lasted for 60 seconds. The mass of the fertilizer discharged at the end of a test run was weighed to determine the discharge mass rate. The uniformity within the values of the discharge mass rate was assessed by the coefficient of variation using the following:

$$CV = \frac{SD}{\bar{X}} * 100 \tag{3}$$

where $CV$ is the coefficient of variation; $\bar{X}$ is the mean and SD is the standard deviation, which are calculated by:

$$\bar{X} = \frac{\sum_{i=1}^{n} X_i}{n} \tag{4}$$

$$SD = \sqrt{\frac{\sum_{i=1}^{n} (X_i - \bar{X})^2}{n-1}} \tag{5}$$

where $X_i$ represents the mass of fertilizer discharge per time ($gs^{-1}$) and n is the number of measurements.

Another main performance indicator used to validate and compare the performance of the two discharge wheels was the velocity of fertilizer falling through the discharge spout. This test lasted for 60 seconds and was run simultaneously with the discharge mass rate and uniformity test. The particle falling velocity was captured using an L (light) model VEO-410 high-speed phantom camera. This camera was operated in single channel mode and allowed the capture of up to 15 images per second with a resolution of over one million pixels. The collected images were processed using the single image analysis method to analyze the velocity of fertilizer particles. In this method, the trajectory of a fertilizer granule was observed at a given time, using the phantom camera control application software version 3.1. This method has been proven effective for analyzing the discharge velocity of fertilizer particles [3, 25].

## Model simulation of fertilizer discharge process

The simulation of the fertilizer discharge process was run for the two discharge wheels, the spiral grooved-wheel and the straight grooved-wheel (Fig 5). The speed of the wheels was varied as 10, 20, 30, 40, 50, and 60 rpm in the simulations. This speed range was within the practical range of 10 to 70 rpm [6].

The discharge mass rate of the two discharge devices was recorded at every rotational speed. The particle falling velocity through the discharge spout at the speed of 37.5 rpm was

**Table 2. The DEM parameters of the materials.**

| Material parameter | Density (kg m$^{-3}$) | Poison ratio | Shear elastic modulus (Pa) |
|---|---|---|---|
| Compound fertilizer | 1437 | 0.25 | 1.25 x 10$^6$[5, 6] |
| Urea fertilizer | 1337 | 0.25 | 2.80 x 10$^7$[22] |
| Discharge assembly | 7850 | 0.29 | 2.05 x 10$^{11}$[23] |

**Table 3. The DEM parameters of contacts between materials.**

| Contact Parameter | Coeff. of restitution | Static friction Coeff. | Rolling friction coeff |
|---|---|---|---|
| Particle to particle of compound fertilizer | 0.307 | 0.372 | 0.123[6] |
| Particle to particle of urea fertilizer | 0.323 | 0.426 | 0.123[6] |
| Compound fertilizer particle to discharge assemble | 0.423 | 0.219 | 0.095[6] |
| Urea fertilizer particle to discharge assemble | 0.318 | 0.205 | 0.168[5] |

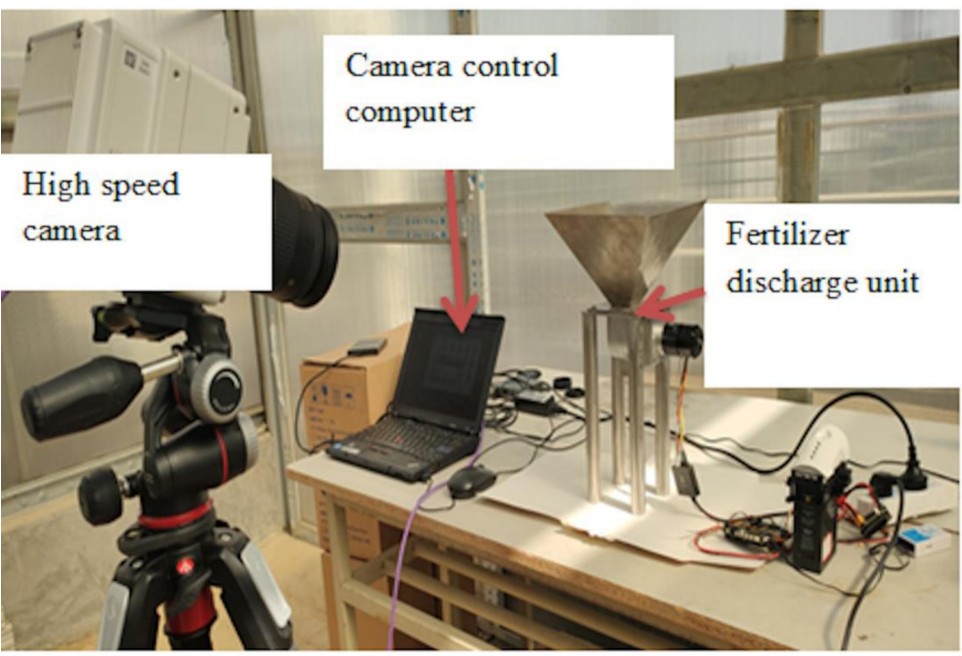

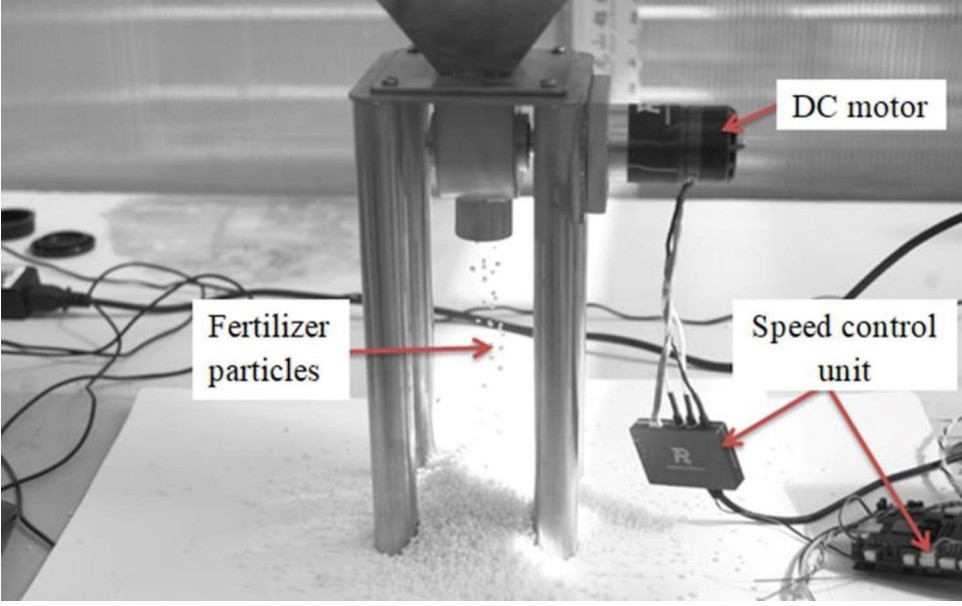

**Fig 4.** (a) Experimental setup; (b) Screenshot of the fertilizer discharge process.

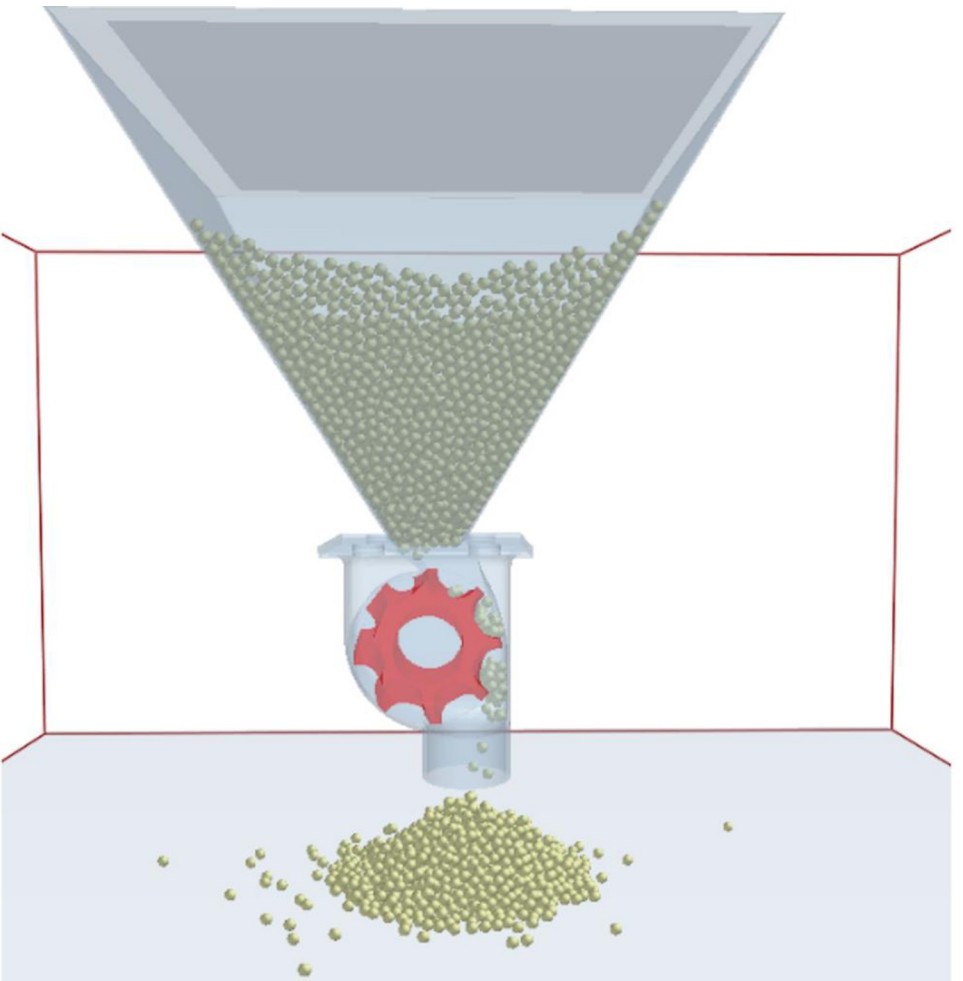

**Fig 5. Screenshots of simulation of the fertilizer discharge process.**

also recorded. In the simulation of fertilizer discharge uniformity, the rotational speed of the discharge wheels was also kept constant at 37.5 rpm. The simulation was replicated 5 times. Each simulation lasted for 10 s, and the mass of fertilizer discharged was recorded. The uniformity of fertilizer discharged was then assessed by the mean, standard deviation, and coefficient of variation using the Eqs (3), (4), and (5).

**Theoretical fertilizer discharge.**   The theoretical fertilizer discharge is calculated from the relationships of the fertilizer discharge quantity $q$ when the grooved-wheel revolves for one revolution with cross-sectional area $A$ of the groove, groove number $Z$, effective working length $L$ of the grooved-wheel, average thickness $C_n$ of the fertilizer driving layer, and revolving speed $n$ of the fertilizer discharging shaft [5], as follows:

$$q = \pi dL\sigma\left(\frac{\alpha(n)f}{t} + C_n\right) \tag{6}$$

where $d$ is the diameter of the grooved-wheel; $\alpha(n)$ is the fertilizer filling coefficient in the groove, which is related to revolving speed; $f$ is the end face area of the groove; $\sigma$ is fertilizer density; $t$ is the pitch between groove teeth.

## Results

### Test results

**Fertilizer discharge mass rate and uniformity of the two discharge wheels.** The spiral grooved-wheel and straight grooved-wheel were compared in terms of fertilizer discharge mass rate and uniformity. For the compound fertilizer, the average discharge mass rate produced by the spiral grooved-wheel was 11.6% higher than that of the straight grooved-wheel (Fig 6A). For the urea fertilizer, a 13.0% higher discharge mass rate was found for the spiral grooved-wheel (Fig 6B).

The test results demonstrated that the coefficients of variation (CV) of the discharge mass rate recorded under the spiral grooved-wheel were as low as 1.9% and 2.6% for the compound and urea fertilizer, respectively. In contrast, the coefficients of variation of the discharge mass rate recorded for the straight grooved-wheel was significantly higher than that of the spiral grooved-wheel, 18% for the compound fertilizer and 17.2% for the urea fertilizer. The results indicated that the fertilizer discharge mass rate of the spiral grooved-wheel was higher and more uniform; it can be a solution to solve the low discharge mass rate and poor fertilizer discharge uniformity problems of the conventional straight grooved-wheel.

**Fertilizer particle falling velocity of the two discharge wheels.** Test results of particle falling velocity were compared between the two discharge devices. For the spiral grooved-wheel, the average velocities of compound and urea fertilizer particles passing through the fertilizer spout were 1.2 m s$^{-1}$ and 1.1 m s$^{-1}$, respectively (Fig 7(A) and 7(B)). These velocities were higher than those recorded for the straight grooved-wheel, which were 0.8 m s-1 for the compound fertilizer and 0.9 m s-1 for the urea fertilizer. The high velocity of fertilizer particles passing through the discharge spout under the spiral grooved-wheel condition can prevent the fertilizer from being bonded in the discharge box and solve the fertilizer blockage problem reported for the conventional straight grooved-wheel.

### Simulation results

**Influence of wheel speed on fertilizer discharge mass rate.** The discharge process was simulated for different rotation speeds, varying from 10 to 60 rpm at a 10 rpm interval. It was

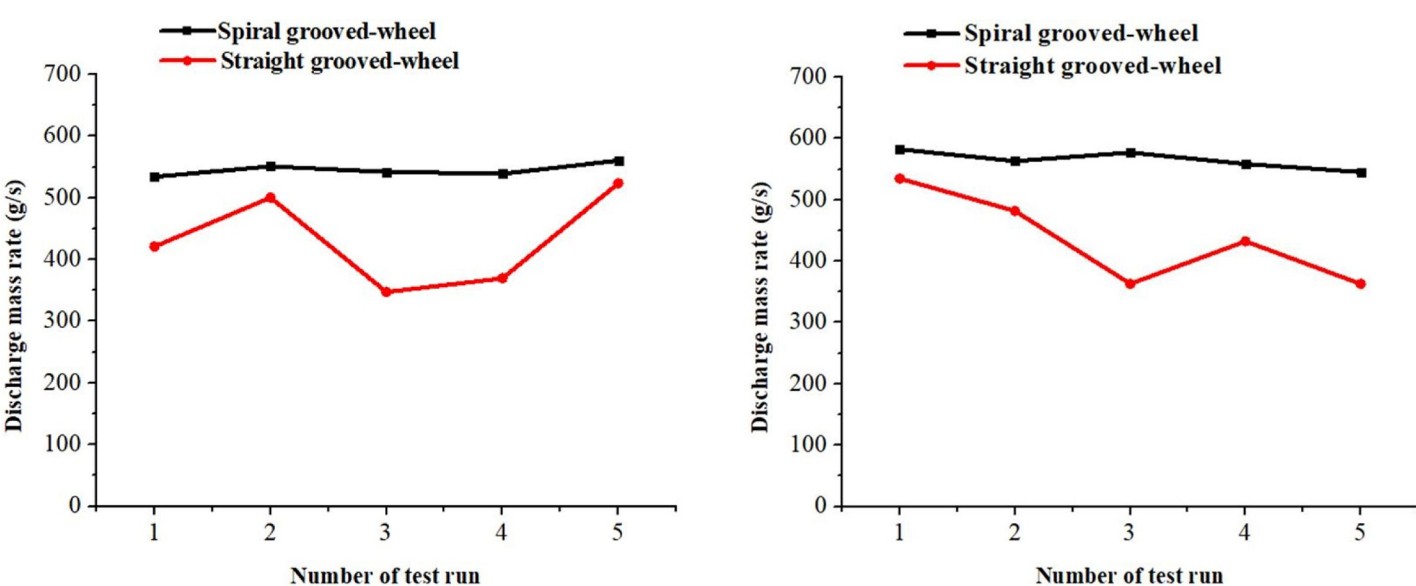

**Fig 6.** Measured fertilizer discharge mass rates of the two discharge devices at constant speed for the (a) compound and (b) urea fertilizer.

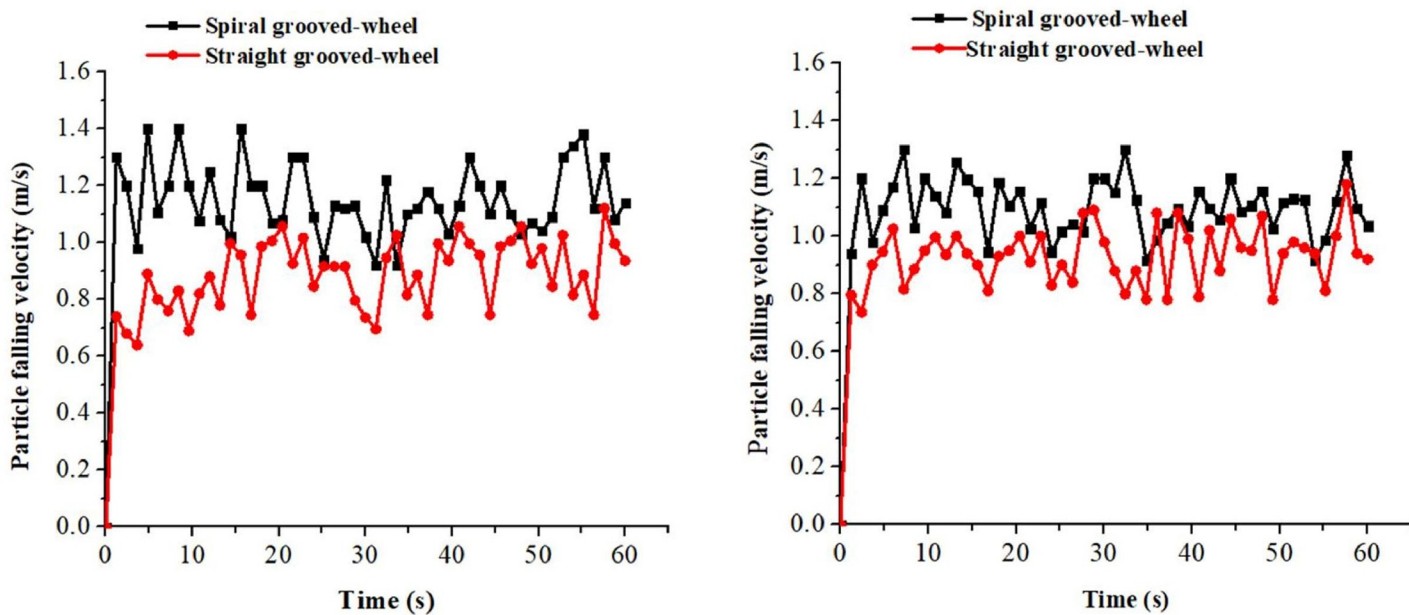

**Fig 7.** Measured fertilizer particle falling velocity of the two discharge devices for the (a) compound and (b) urea fertilizer.

evident that the rotation speed has a large influence on the fertilizer discharge mass rate for the two discharge wheels (Fig 8(A) and 8(B)). The fertilizer discharge mass rate increases as the rotation speed increases. The fertilizer discharged mass rate of the spiral grooved-wheel was higher than that of the straight grooved-wheel at all the rotation speeds examined, and the difference in fertilizer discharged mass rate was more pronounced when using the compound fertilizer (Fig 8A). Therefore, the spiral grooved-wheel designed is more favorable to the discharge of both fertilizers.

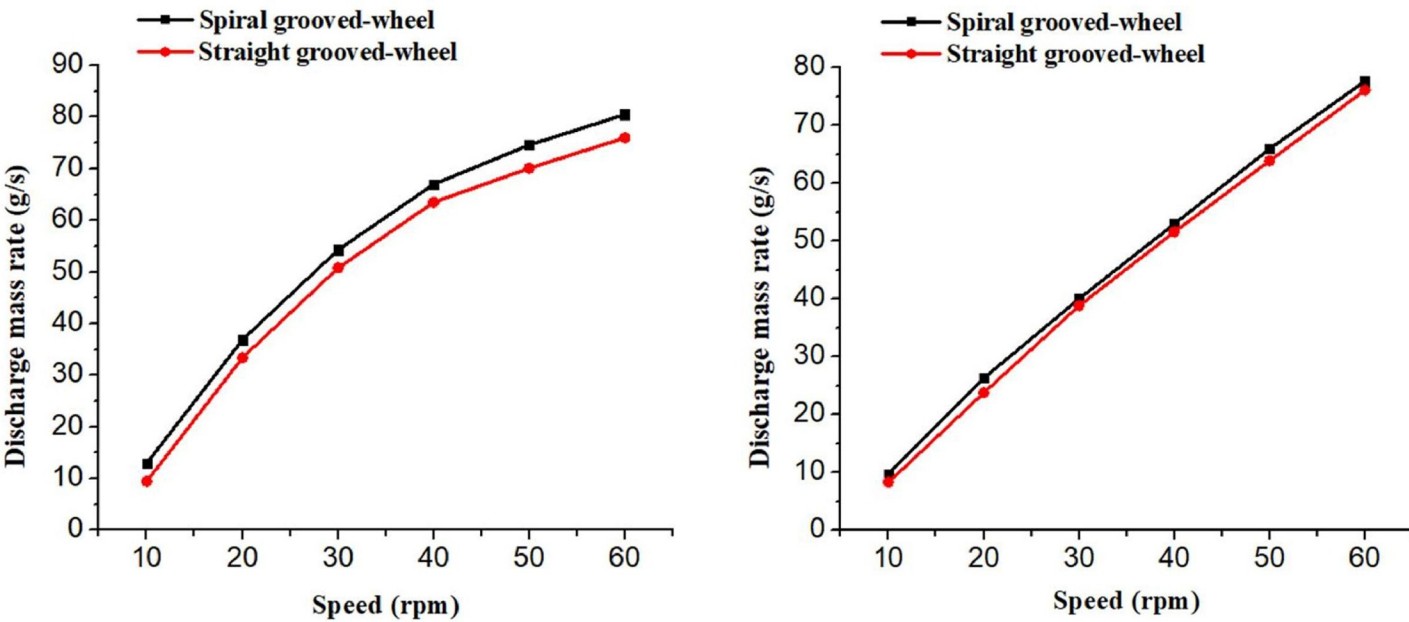

**Fig 8.** Simulated discharge mass rates of the two discharge devices for the (a) compound and (b) urea fertilizer.

**Influence of rotation speed on the particle falling velocity of fertilizer granules.** From the simulation, it was observed that the average particle falling velocity for the compound fertilizer was about 1.08 m s-1 under the spiral grooved-wheel, and that of the straight grooved-wheel model was 0.99 m s$^{-1}$ (Fig 9A). Similarly, the average particle falling velocity for the urea fertilizer was higher with the spiral grooved-wheel (1.2 m s$^{-1}$) than with the straight grooved-wheel (1.1 m s$^{-1}$) (Fig 9B). This result demonstrated that the fertilizer granules discharged by the spiral grooved-wheel flow faster. In addition, the continuity of fertilizer discharged by the straight grooved-wheel was lower than that of the spiral grooved-wheel. This further verifies the improved performance of the fertilizer applicator when using the spiral grooved-wheel designed.

**Fertilizer discharge uniformity.** The fertilizer discharge uniformity of the spiral grooved-wheel was compared with the conventional straight grooved-wheel at a constant wheel rotation speed (37.5 rpm) using the DEM simulation results. For both types of fertilizers and both discharge devices, the discharge mass rate varies among the simulation runs (Fig 10(A) and 10 (B)). However, the mass rate of the spiral grooved-wheel does not vary as much as that of the straight grooved-wheel. Thus, the former had better stability in terms of performance in this regard.

According to the Eq (3), the coefficients of variation (CV) of fertilizer mass discharged by the spiral grooved-wheel were 5.1% and 7.02% for the compound and urea fertilizer, respectively. Whereas, the coefficients of variation for the straight grooved-wheel were significantly higher (25.6% and 24.2% for the compound and urea fertilizer, respectively). This indicated that the designed discharge spiral grooved-wheel is more stable and favorable for uniform fertilizer applications compared to that of the straight one. This can solve the problem of poor stability and non-uniform fertilizer application reported for the conventional discharge devices.

## Comparisons between simulations and measurements

**Comparison of fertilizer discharge mass rate and uniformity.** For validation, test results were compared with the simulation results at the same wheel rotation speed. The agreement

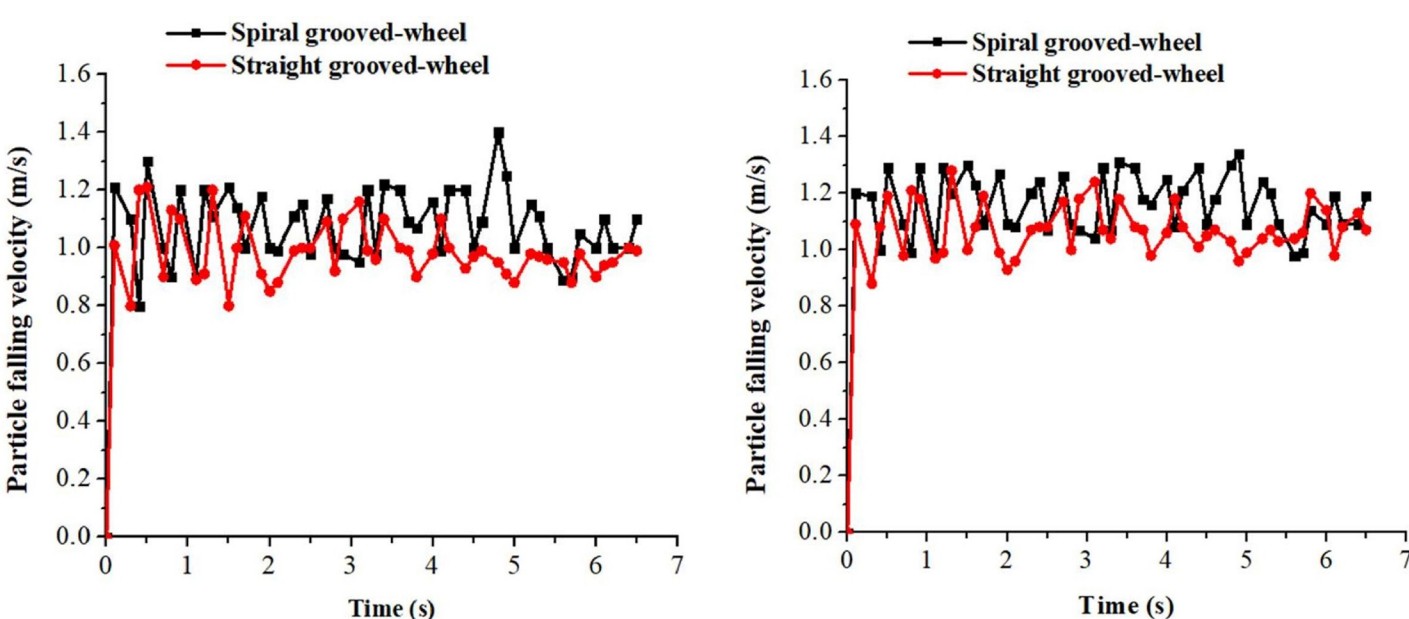

**Fig 9.** Simulated fertilizer particle falling velocities of the two discharge devices for the (a) compound and (b) urea fertilizer.

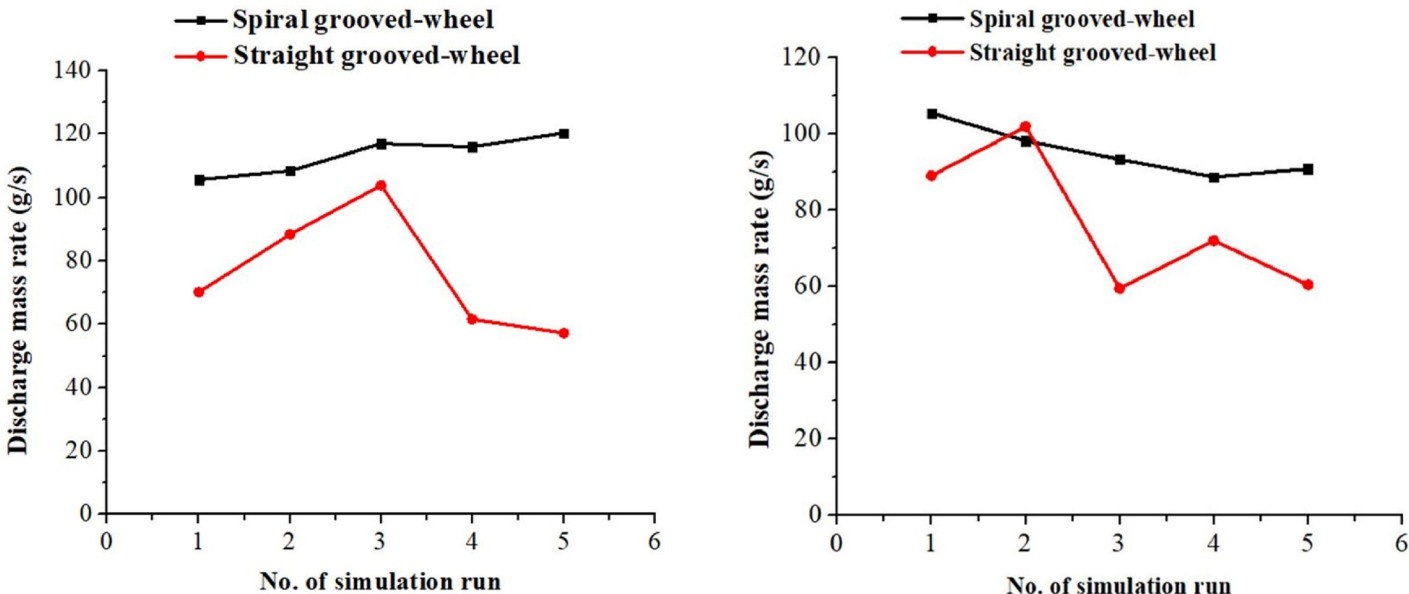

**Fig 10.** Simulated fertilizer discharge rates of the two discharge devices for the (a) compound and (b) Urea fertilizer.

between the two sets of results was assessed with relative errors. Relative error was defined as the percentage of the absolute difference between measured and simulated values over the measured value. The simulated and measured values of fertilizer discharge mass rate and discharge uniformity were very much comparable under the spiral grooved-wheel (Table 4). Over the five simulation runs, the average relative errors for the compound and urea fertilizers were all less than 10%. The two sets of results of fertilizer discharge mass rate recorded under the straight grooved-wheel did not agree very well as indicated by the average relative errors both greater than 20% (Table 4). The variation coefficients of fertilizer discharge uniformity of the two discharge devices were also compared. The results show that the average coefficients of variation of fertilizer discharge uniformity of the spiral grooved-wheel were low (< 10%) compared to the 20% that was recorded under the straight grooved-wheel.

For both types of fertilizers, the overall relative errors between the simulations and measurements were quite low for the spiral grooved-wheel; this demonstrated that the DEM model captured better the actual behavior and performance, in terms of discharge mass rate and uniformity, of the spiral grooved-wheel than the conventional straight grooved-wheel.

**Comparison of fertilizer falling velocity.** Simulated and measured values of particle falling velocity were also compared under the same wheel speed. From Table 5, it is evident that

**Table 4. Comparison of measured and simulated values of fertilizer discharge mass rate for the two discharge wheels.**

| | Spiral grooved-wheel | | Straight grooved-wheel | |
| | Relative error (%) | | Relative error (%) | |
| Run no. | Compound fertilizer | Urea fertilizer | Compound fertilizer | Urea fertilizer |
| --- | --- | --- | --- | --- |
| 1 | 1.81 | 8.08 | 23.08 | 23.08 |
| 2 | 21.39 | 7.33 | 18.47 | 8.07 |
| 3 | 2.58 | 20.18 | 1.09 | 24.35 |
| 4 | 0.77 | 3.79 | 23.08 | 23.08 |
| 5 | 6.19 | 2.37 | 49.57 | 23.08 |
| Mean | 6.55 | 8.35 | 23.06 | 20.33 |

Table 5. Comparison of measured and simulated values of fertilizer falling velocity.

| Fertilizer | Spiral grooved-wheel | Straight grooved-wheel |
|---|---|---|
| | Relative error (%) | Relative error (%) |
| Compound fertilizer | 9.8 | 19.2 |
| Urea fertilizer | 8.3 | 18.2 |

the average relative errors between the simulated and measured values for the compound and urea fertilizers recorded under the spiral grooved-wheel were significantly lower than those recorded with the straight grooved-wheel. This proves the feasibility and effectiveness of DEM analysis on the operation of the spiral grooved-wheel fertilization device designed in this study.

## Discussion

In small scale farming, improved control and precision of fertilizer discharge rate are particularly important in fertilizer management of the crop cycle because fertilizers are expensive and are applied in a manner that largely determines the final crop yield [26–30]. Improper design of fertilizer discharge devices can result in non-uniform application rates, causing fertilizer losses of up to 40% [31, 32].

In this study, a spiral grooved-wheel fertilizer discharge device was developed. It is simple to operate, has a stable discharge rate, and can be used for small and medium-size farms. The performance of the spiral grooved-wheel was compared with the conventional straight grooved-wheel by measuring the discharge mass rate, discharge uniformity, and particle falling velocity. Both the test and DEM simulation results showed that in general, the spiral grooved-wheel outperformed the straight grooved-wheel. A similar result was also reported by Liping et al. [5], and they considered the spiral grooved-wheel helix angle as the major influencing factor. Maleki et al. [10] and Kara et al. [33] also found that fertilizer discharge mass rate and uniformity of the spiral grooved-wheel fertilizer discharge device was the best among others. The fertilizer falling velocity of the two discharge wheels monitored showed that the average velocity of fertilizer particles falling through the fertilizer discharge spout was higher under the spiral grooved-wheel for both the two types of fertilizers used in this study. This result reaffirms that the spiral grooved-wheel designed in this study is more conducive for the discharge of different sizes of fertilizer granules as reported in the literature [5, 11, 12]; also, in this case, the fertilizer particle discharge by the spiral grooved-wheel will be faster than that of the straight grooved-wheel. This is important to solve the problem of poor stability, blockage and non-uniform fertilizer application challenges caused by the conventional straight-grooved wheels previously reported by other researchers [7–9]. The spiral grooved-wheel designed in this study worked well in the rotation speeds between 30 and 45 rpm. Within this range, DEM simulated and measured values were not significantly different. The relative errors were less than 10% for all the discharge mass rates, discharge uniformity, and particle falling velocity. This demonstrated that our proposed model can simulate the working process of the spiral grooved-wheel with reasonably good accuracy. Further development of the spiral grooved-wheel could significantly improve the control over the fertilizer inputs of small farms, reduce labor costs and at the same, enabling them to increase productivity. Our simulation approach also provides useful references for further optimization of the design parameters of the spiral grooved-wheel fertilizer discharge machines and for developing innovative fertilizer application technologies.

## Conclusions

The results of this study indicated that the DEM model developed can be used to evaluate the discharge performance of the spiral grooved-wheel, and provide reference values for the design of spiral grooved-wheel fertilizer application devices. Comparison of the DEM simulation and measured results revealed a good agreement in general and consistent variation trends. The results can be used to improve the conventional fertilizer application methods, and they are important for the improvement of fertilizer discharge mass rate and uniformity. This new method delivers fertilizer in a more precise amount and position; provide a sustainable and efficient use of chemical fertilizer for small and medium farms. Future work will focus on adopting the developed spiral grooved-wheel device on a two-row rice transplanter to provide stable, consistent, and accurate fertilizer application rates in the field.

## Supporting information

**S1 File.**
(PDF)

## Author Contributions

**Conceptualization:** Kemoh Bangura, Hao Gong, Long Qi.

**Data curation:** Kemoh Bangura, Hao Gong, Yinghu Cai.

**Formal analysis:** Kemoh Bangura, Hao Gong.

**Funding acquisition:** Long Qi.

**Investigation:** Kemoh Bangura, Long Qi.

**Methodology:** Kemoh Bangura, Hao Gong, Kaifeng Liao, Long Qi.

**Project administration:** Kemoh Bangura, Hao Gong, Long Qi.

**Resources:** Kemoh Bangura, Long Qi.

**Software:** Kemoh Bangura, Hao Gong.

**Supervision:** Jinwei Liu, Long Qi.

**Validation:** Kemoh Bangura, Ruoling Deng, Ming Tao, Chuang Liu, Yinghu Cai, Kaifeng Liao.

**Visualization:** Kemoh Bangura, Long Qi.

**Writing – original draft:** Kemoh Bangura.

**Writing – review & editing:** Kemoh Bangura, Long Qi.

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
