## [Decision Letter · Decision Letter 0]

8 Jun 2020

PONE-D-20-13766

Simulation analysis of fertilizer discharge process using the discrete element method (DEM)

PLOS ONE

Dear Dr. Long,

Thank you for submitting your manuscript to PLOS ONE. After careful consideration, we feel that it has merit but does not fully meet PLOS ONE’s publication criteria as it currently stands. Therefore, we invite you to submit a revised version of the manuscript that addresses the points raised during the review process.

We look forward to receiving your revised manuscript.

Kind regards,

Fang-Bao Tian

Academic Editor

PLOS ONE

Journal Requirements:

Reviewers' comments:

Reviewer's Responses to Questions

**Comments to the Author**

1. Is the manuscript technically sound, and do the data support the conclusions?

Reviewer #1: Yes

2. Has the statistical analysis been performed appropriately and rigorously? 

Reviewer #1: Yes

3. Have the authors made all data underlying the findings in their manuscript fully available?

Reviewer #1: Yes

4. Is the manuscript presented in an intelligible fashion and written in standard English?

Reviewer #1: Yes

5. Review Comments to the Author

Reviewer #1: The manuscript’s topic deals with the testing and simulation of fertilizer discharge process of a newly developed discharge device versus a conventional one. The topic of the manuscript falls within the general scope of the journal and contains potentially useful information for readers. The use of the simulation method of discrete element method is innovative. Generally speaking, the manuscript is acceptable for publication after some minor revisions.

1. Line 79, spell out ‘Fig’ at the beginning of the sentence

2. Tables, please remove all vertical lines and some of unnecessary horizontal lines, refer to the journal guidance

3. Tables 2 and 3, please specify the sources of each DEM parameter specifically in the tables

4. Line 140, ‘given the running time of 60 seconds’ redundant information, suggest deleting

5. Line 156, please provide the brand and model of the camera used

6. Line 160, please explain what is ‘application 3.1’

7. How to calculate the theoretical fertilizer discharge?

8. In actual operation, the fertilizer distributor has a forward movement speed. How does it affect the parameter information simulated in the article?

9. What is the basis for selecting the height parameter of equipment setting?

6. PLOS authors have the option to publish the peer review history of their article (what does this mean?). If published, this will include your full peer review and any attached files.

Reviewer #1: No

---

## [Author Response · Author response to Decision Letter 0]

22 Jun 2020

June 18, 2020

Fang-Bao Tian 

Academic Editor

PLOS ONE

PONE-D-20-13766

Dear Dr. Tian,

Thank you very much for reviewing our manuscript. We also greatly appreciate the reviewer for his complimentary comments and suggestions. We have addressed the reviewer’s comments and suggestions and revised the manuscript accordingly. We have ensured that the manuscript meets PLOS ONE's style requirements and the corresponding author has linked his ORCID account in PLOS ONE Editorial Manager System. 

Please find attached a point-by-point response to reviewer’s concerns. We hope that you find our responses satisfactory, and that the manuscript is now acceptable for publication.

Sincerely,

Long Qi (Ph.D),

Professor

South China Agricultural University

NO. 483, Wushan, Tianhe, CHINA

Email: qilong@scau.edu.cn

Response to Reviewer

We appreciate the reviewer’s comments. The followings are our point-by-point responses:

1. Line 79, spell out ‘Fig’ at the beginning of the sentence

Response: Fig has been spelt out in line 82

2. Tables, please remove all vertical lines and some of unnecessary horizontal lines, refer to the journal guidance

Response: We corrected the tables as indicated

3. Tables 2 and 3, please specify the sources of each DEM parameter specifically in the tables

Response: We have specified the source of each DEM parameters in tables 2 and 3.

4. Line 140, ‘given the running time of 60 seconds’ redundant information, suggest deleting

Response: The sentence ‘given the running time of 60 seconds’ has been deleted in line 143.

5. Line 156, please provide the brand and model of the camera used

Response: We have provided the brand and model of the camera used in line 159.

6. Line 160, please explain what is ‘application 3.1’

Response: ‘application 3.1’ has been corrected to application software version 3.1 in line 164.

7. How to calculate the theoretical fertilizer discharge?

Response: We have added information to the paper (page 10, lines 181 – 190) to explain how to calculate the theoretical fertilizer discharge.

8. In actual operation, the fertilizer distributor has a forward movement speed. How does it affect the parameter information simulated in the article?

Response: We agree with the reviewer that in actual operation, the fertilizer distributor has a forward movement speed. However, the article was tailored to compare the discharge performance of the conventional straight grooved-wheel and the spiral grooved-wheel through simulations and platform tests. The platform test and simulation results indicated the validity and effectiveness of DEM analysis on the operation of the spiral grooved-wheel fertilization device as demonstrated in the article. Our next research plan as mentioned in the conclusions section of this article (lines 343 – 345) will focus on adopting the developed spiral grooved-wheel fertilizer distributor on a two-row rice transplanter, considering the forward movement speed of the transplanter. The effect of forward movement speed on discharge performance and accurate fertilizer application rates will be assessed. This will further prove the effectiveness of the simulation model in the article.

9. What is the basis for selecting the height parameter of equipment setting?

Response: As requested by the reviewer, we have added four sentences to the paper (lines 73 – 76) to explain the basis for selecting the height of the equipment setting.

---

## [Editor Report · Decision Letter 1]

24 Jun 2020

Simulation analysis of fertilizer discharge process using the discrete element method (DEM)

PONE-D-20-13766R1

Dear Dr. Long,

We’re pleased to inform you that your manuscript has been judged scientifically suitable for publication and will be formally accepted for publication once it meets all outstanding technical requirements.

Kind regards,

Fang-Bao Tian

Academic Editor

PLOS ONE
---

## [Editor Report · Acceptance letter]

26 Jun 2020

PONE-D-20-13766R1 

Simulation analysis of fertilizer discharge process using the discrete element method (DEM) 

Dear Dr. Qi:

I'm pleased to inform you that your manuscript has been deemed suitable for publication in PLOS ONE. Congratulations! Your manuscript is now with our production department. 

Kind regards, 

on behalf of

Dr. Fang-Bao Tian 

Academic Editor

PLOS ONE